# Establishing the safety of waterbirth for mothers and babies: a cohort study with nested qualitative component: the protocol for the POOL study

Rebecca Milton [1], Julia Sanders,[2] Christian Barlow,[1] Peter Brocklehurst,[3] Rebecca Cannings-John,[1] Sue Channon,[1] Christopher Gale [4] Abigail Holmes,[5] Billie Hunter,[2] Shantini Paranjothy,[6] Fiona V Lugg-Widger [1] Sarah Milosevic,[1] Leah Morantz,[7] Rachel Plachcinski,[8] Mary Nolan,[9] Michael Robling [1]

**Correspondence to**
Rebecca Milton;
miltonrl1@cardiff.ac.uk

## ABSTRACT

**Introduction** Approximately 60 000 (9/100) infants are born into water annually in the UK and this is likely to increase. Case reports identified infants with water inhalation or sepsis following birth in water and there is a concern that women giving birth in water may sustain more complex perineal trauma. There have not been studies large enough to show whether waterbirth increases these poor outcomes. The POOL Study (ISRCTN13315580) plans to answer the question about the safety of waterbirths among women who are classified appropriate for midwifery-led intrapartum care.

**Methods and analysis** A cohort study with a nested qualitative component. Objectives will be answered using retrospective and prospective data captured in electronic National Health Service (NHS) maternity and neonatal systems. The qualitative component aims to explore factors influencing pool use and waterbirth; data will be gathered via discussion groups, interviews and case studies of maternity units.

**Ethics and dissemination** The protocol has been approved by NHS Wales Research Ethics Committee (18/WA/0291) the transfer of identifiable data has been approved by Health Research Authority Confidentiality Advisory Group (18CAG0153).
Study findings and innovative methodology will be disseminated through peer-reviewed journals, conferences and events. Results will be of interest to the general public, clinical and policy stakeholders in the UK and will be disseminated accordingly.

## INTRODUCTION

In 1992, the House of Commons Health Committee recommended hospitals should provide women with the use of a birth pool for labour 'where this is practicable'.[1] In the intervening years, the popularity of the use of water immersion for labour and birth in the UK has increased, and since 2007 The National Institute for Health and Care Excellence (NICE) recommends water immersion

## Strengths and limitations of this study

► Using large retrospective and prospective datasets concomitantly provides six years data over a three year study period.
► Ability to look at all neonatal outcomes and treatments across the wide geographical range and number of units.
► Using existing, routine data enhanced by prospective data to investigate the safety of waterbirth across a range of outcomes.
► Data collected will only represent users of Wellbeing Software's EuroKing maternity software system.
► Allocation is not random, so unmeasured confounding is possible.

analgesia be made available to all clinically appropriate, low-risk women in labour.[2]

The Cochrane review of water immersion during labour provided evidence supportive of pool use for labour analgesia but could not answer the question relating to the safety of waterbirth for mother or baby. The review included 12 trials (3243 women), nine of which focused on the first stage of labour. Results from six studies looking at the first stage of labour found a significant reduction in the rate of regional analgesia/anaesthesia among women allocated to water immersion compared with no immersion (478/1254 vs 529/1245, respectively; risk ratio 0.90; 95% CI 0.82 to 0.99).[3]

Many professionals and parents have strong opinions on waterbirth. Some are great advocates, promoting benefits of waterbirth, while others remain concerned that women who give birth in water may be exposing themselves or their baby to additional unnecessary risks.[4–6]

This study is collecting data on births to all women in 26 UK maternity units from 2015 and is identifying the numbers, proportion and characteristics of women who use water immersion during labour or birth. The study will also establish whether waterbirth is as safe for mothers and their infants as using a pool during labour but getting out prior to birth. Data will be collected on 15 000 waterbirths and 15 000 land births among women with uncomplicated pregnancies from National Health Service (NHS) sites which use Wellbeing Software's (WS) maternity software system EuroKing. This study will use data recorded routinely as part of standard maternity care and stored on respective NHS site's servers. For infants admitted to a neonatal unit (NNU), the study will also use data held by the National Neonatal Research Database (NNRD). Data needed to answer some study questions are already recorded, for example, perineal trauma, therefore, data from births from 2015 onwards can be included. Existing, routinely collected data does not capture all information to answer the study questions, data items missing from existing routinely collected data include infants receiving antibiotics without admission to NNU and whether following waterbirth the placenta is delivered in or out of water. Cardiff University (CU) has worked with WS in order to develop study specific data fields which will enable these data to be captured in participating sites from 2019 onwards.

## METHODS AND ANALYSIS
### Primary research objective
To establish whether for low-risk[2] women who use a pool during labour, waterbirth, compared with leaving a pool prior to birth, is as safe for mothers and infants.

### Secondary objectives
To establish:
1. Overall proportion and characteristics of women who use a pool for labour or birth, compared with those who do not use a pool.
2. Characteristics of, and outcomes for, women with risk factors, who use a pool during labour.
3. Characteristics of, and outcomes for, women who develop labour complications who use a pool during labour.
4. Factors associated with rates of pool use in individual maternity units.

### Primary outcomes
#### Maternal primary outcome
Obstetric Anal Sphincter Injury (OASIS).

#### Infant primary outcome
A composite of 'adverse infant outcomes or treatment' to include:
1. Any NNU admission requiring respiratory support.
2. Antibiotic administration within 48 hours of birth (with/without culture proven infection).

3. Intrapartum stillbirth or all deaths prior to NNU/post-natal ward discharge.

### Secondary outcomes
#### Maternal secondary outcomes
Maternal intrapartum:
► Shoulder dystocia and required management.
► Management of the third stage of labour (whether the placenta was intended to be, or delivered in or out of water).
► Obstetric involvement in care.
► Incidence and management of perineal trauma.
► Maternal position at birth.
  Maternal postnatal:
► Duration of postnatal stay.
► Breastfeeding.
► Need for higher-level care.
► Maternal readmission to hospital within 7 days of birth.

#### Infant secondary outcomes
► Timing of cord clamping.
► Apgar scores (1, 5 and 10 min).
► Cause of intrapartum stillbirth or death prior to NNU/postnatal ward discharge.
  Incidence of:
► NNU admission requiring respiratory support.
► Antibiotic administration within 48 hours of birth (with/without culture proven infection).
► Intrapartum stillbirth or neonatal death prior to NNU/postnatal ward discharge occurring within 7 days of birth.
► Neonatal resuscitation.
► Snapped umbilical cord prior to clamping.
► Skin-to-skin contact at birth.
► First breastfeed within first hour.
► Culture proven infection.
► Brachial plexus injury.
► Treatment for jaundice.
► Readmission to hospital within 7 days of birth.
► Receipt of therapeutic hypothermia.
► NNU admissions.
► Respiratory support.

A further set of secondary outcomes were piloted at one site including highest C reactive protein (CRP) results and successful/attempted lumbar puncture. Data collection was successful and included in the dataset for all sites.

### Study design
A natural experiment using a cohort design with a nested qualitative component will answer study objectives using a combination of retrospective and prospective data in electronic NHS maternity and neonatal information systems. The qualitative component will explore factors influencing pool use and waterbirth. CU has partnered with WS who supply EuroKing maternity software system in the UK and the NNRD to link data on infants transferred to NNU.

To answer all objectives approximately 600 000 individual computerised maternity records held on secure NHS servers in 26 NHS sites from January 2015 will be accessed. To provide necessary denominator data and to be able to compare characteristics of pool/non-pool users, a dataset will be extracted relating to women who did not use a pool in labour, a more extensive dataset will be extracted for women who used a pool in labour. An important clinical question is whether there is a differential effect of waterbirth on severe perineal trauma among nulliparous and parous women. A larger sample size is required for the maternal (30 000) compared with the infant (16 200) primary outcome. To inform the maternal primary outcome, severe perineal trauma, which is already collected in the maternity information system at study sites, data will be extracted relating to births between 1 January 2015 to the end of data collection. The infant composite primary outcome includes data items added at site opening, as essential data are not collected routinely. For this reason, data used to inform the infant primary outcome will only include births occurring between the date of an individual site opening and the end of data collection.

NNRD[7 8] holds individual patient-level data on all infants admitted for NHS neonatal care in England, Wales and Scotland. To obtain detailed treatment and outcome information on infants admitted to NNU, following their mothers' pool use in labour, the identifiers during the

period of prospective data collection will be extracted and matched to any records held by the NNRD.

The primary study aim is to compare maternal and infant outcomes for low-risk women who gave birth in water (group 1) against low-risk women who left the water prior to birth, with no risk-based or clinical reasons (group 2). Figure 1 shows and details the study population groups. Women classified as low risk for study purposes will be at term ($37+^0$–$41+^6$ weeks gestation), with a singleton fetus in spontaneous labour with an absence of factors that indicate that obstetric or other medical staff should have involvement in her care, or birth in an obstetric unit is advised.[2]

### Data providers and datasets

To answer the research questions, two datasets will be used, data extracted from EuroKing maternity software system and data held by the NNRD. EuroKing forms a comprehensive clinical dataset and is currently used by 26 of the maternity NHS Trusts and Health Boards in the UK. All 200 NNUs in England, Wales and Scotland form the UK Neonatal Collaborative and contribute electronic health record data to the NNRD. The NNRD is a national resource formed of the Neonatal Dataset (an NHS Information Standard), comprising of 450 clearly defined variables extracted at patient level from the commercial electronic health record used by all UK NNUs. For the purpose of the POOL study, NNRD data will only be used

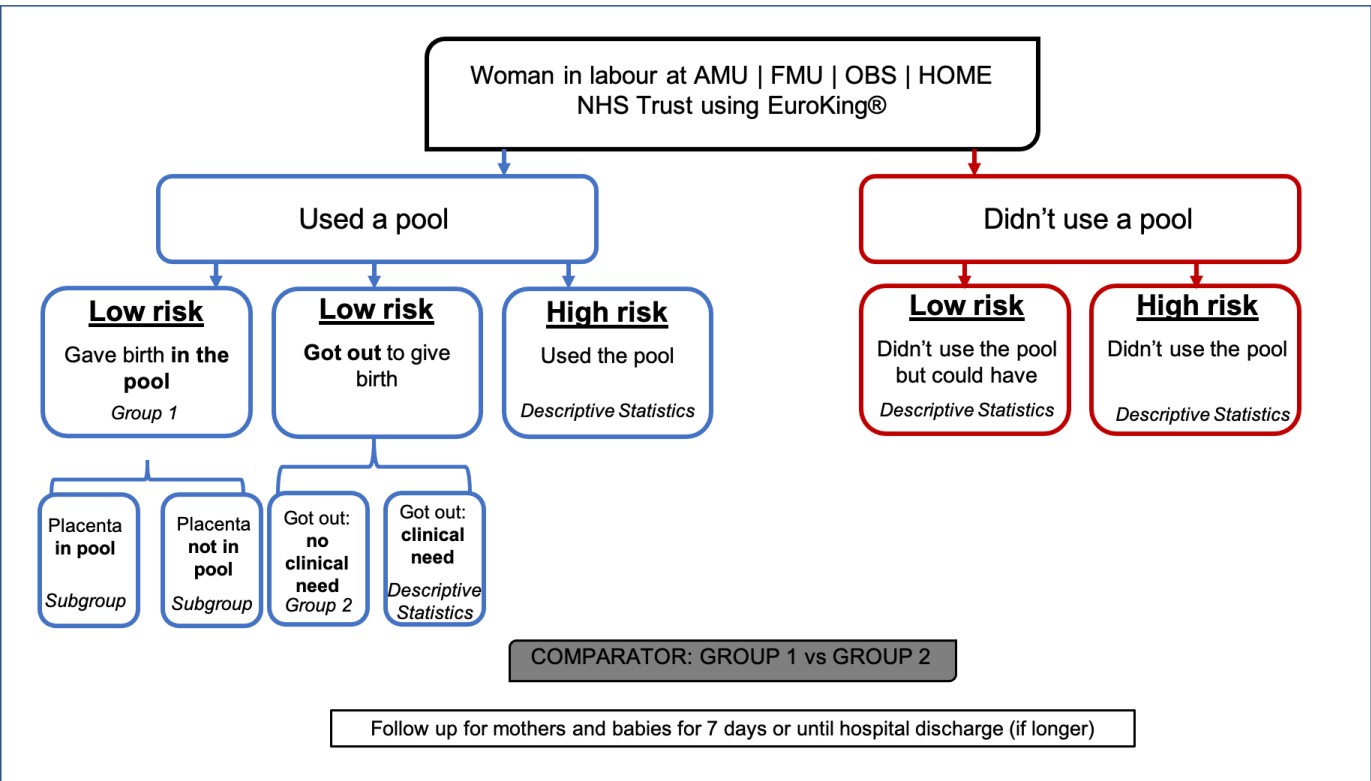

**Figure 1** Study population groups: overview of the four groups of women within the POOL study population and how these groups will be compared and their data reported. AMU, Alongside Midwifery Unit; FMU, Freestanding Midwifery Unit; OBS, Obstetric Unit; NHS, National Health Service.

from England and Wales, as no units in Scotland are supported by WS.

For the retrospective data collection, the data extract will be created by WS for the period January 2015 until prospective data collection commences (site opening). This extract will be created remotely by WS accessing the participating site's server. A unique study number will be generated prior to data leaving the study sites. WS will transfer a pseudonymised version of this extract to CU and a separate extract of data containing the unique study number, and identifiers to NNRD (pool use) using a secure file transfer process. NNRD will proceed to match the data received from WS to ensure complete records are obtained for infants transferred to NNU and send the pseudonymised dataset to CU.

For prospective data, the same format will be followed as for the retrospective data. Prospective data extracts will include the new variables added to EuroKing (overview of key data items in table 1). A separate syntax will direct the NHS number, unique study number and other identifiers of infants born to all women who used a pool during labour, after site opening, to the NNRD.

NNRD will send three matched datasets to CU for analysis (pilot study, once all sites are opened and at the end

| Table 1 | Data sources | | |
| --- | --- | --- | --- |
| **Data provider** | **Data source** | **Data collection period** | **Key data items** |
| Wellbeing Software | NHS site (Maternity unit) | From January 2015 | Routinely collected data items:<br>► Demographics (including parity, age, ethnicity and deprivation)<br>► Use of pool for labour analgesia/waterbirth<br>► Maternal health<br>► Labour<br>► Birth<br>► Pregnancy history<br>► Maternal medical and obstetric history<br>► Midwifery or obstetric led intrapartum care<br>► Delayed cord clamping (>60s after birth)<br>► Type of intended care throughout labour<br>► Maternal health conditions |
| Wellbeing Software | NHS Site (Maternity Unit) | From site opening | ► Risk status at pool entry<br>► Reasons for pool exit<br>► Obstetric care or input prior to birth<br>► Birthing complications<br>► Cord snapping prior to clamping<br>► Obstetric care in immediate postnatal period.<br>► Syntocinon administered in water for labour augmentation<br>► Cardiotocography used in water<br>► Placenta delivery in or out of water (waterbirths only)<br>► Infant antibiotic administration<br>► Infant lumbar puncture<br>► Infant blood culture<br>► Highest neonatal CRP result<br>► Treatment for jaundice |
| National Neonatal Research Database (Data relating to neonates admitted to a neonatal unit) | NHS Site (Neonatal Unit) | From January 2015 | ► Neonatal unit admission and transfer<br>► Level of care and number of days received<br>► Respiratory support<br>► Intravenous antibiotic administration<br>► Intrapartum stillbirth or infant death prior to NNU/postnatal ward discharge<br>► Timing of cord clamping<br>► Apgar scores<br>► Resuscitation at birth<br>► Culture proven infection<br>► Brachial plexus injury<br>► Treatment for jaundice<br>► Readmission to neonatal unit within 7 days of birth<br>► Therapeutic hypothermia |

CRP, C reactive protein; NHS, National Health Service; NNU, neonatal unit.

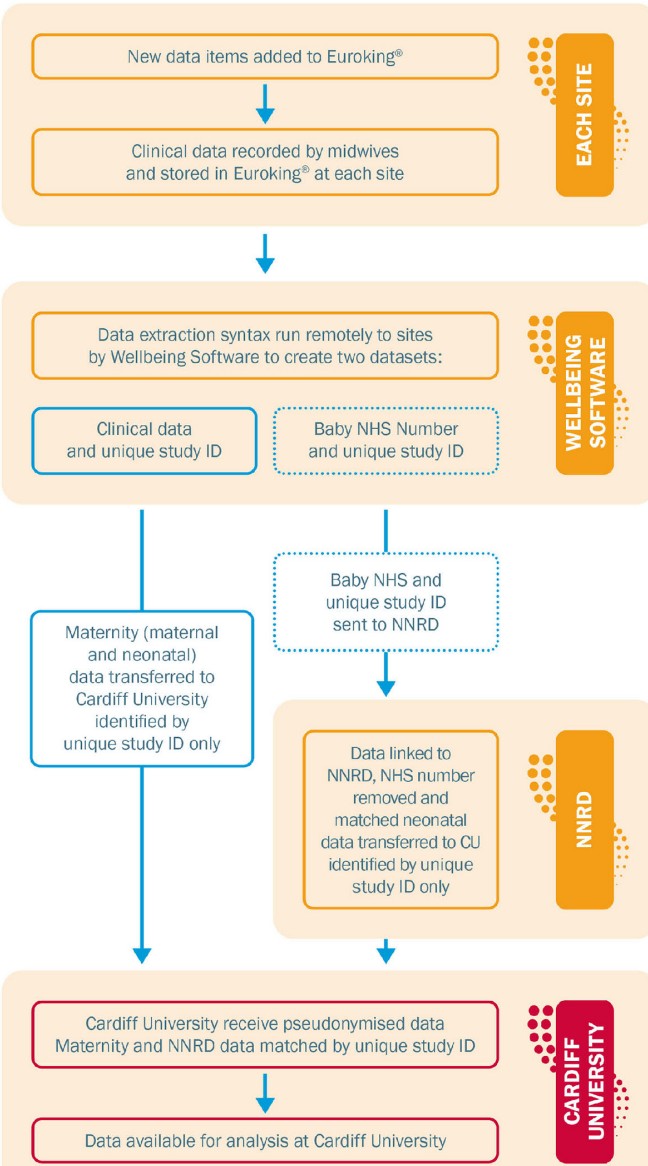

**Figure 2** Data flow: a description of how the data flows from new variables being input into the EuroKing maternity system to analysis. NHS, National Health Service; NNRD, National Neonatal Research Database.

of study). Any NNRD data describing infants matched to the study will have NHS numbers removed prior to data being transferred back to CU with the unique study number.

Use of this method of case labelling will enable CU only to hold pseudonymised data, while facilitating the identification of mother/infant dyads and enable the matching of the NNU admission record onto to the mother and infant record held in EuroKing (figure 2).

## Opportunity to Opt-Out

This study will use data collected in NHS electronic systems that will be pseudonymised prior to transfer to the CU study team (table 1). Approval for the transfer of identifiable patient data from WS to NNRD in order to match the infants transferred to NNU has been obtained under Section 251 (s251) of the NHS Act 2006.

Participants will have the option to opt-out by informing the maternity unit that they do not wish to participate. CU will provide individual sites with patient information leaflets, posters and take-home cards. Individual sites are responsible for ensuring all women are provided with relevant information to ensure they are aware of the option to opt-out.

## Qualitative component

The aim of the qualitative component of the study is to identify and explore factors which influence the use of birth pools and giving birth in water.

Phase 1:

Six closed online discussion groups hosted on the CU website and telephone interviews:
1. Women with recent experience of maternity services.
2. Heads of midwifery and midwifery managers.
3. Consultant midwives.
4. Band 5/6 clinically focused midwives.
5. Obstetricians.
6. Neonatologists and paediatricians.

Phase 2:

In-depth organisational case studies across three study sites, comprising midwifery-led and obstetric units with a range of waterbirth rates. Data points:
► Key documents, online information and existing data relating to pool use and waterbirth.
► Information relating to unit equipment and facilities.
► Group or individual discussions with staff and lay representatives.

## Study participants inclusion and exclusion criteria

Cohort study:

Main analysis: All women at low-risk of complications who use a birth-pool or bath for water immersion during established labour between 1 January 2015 and the end of data collection. Women will be classified as 'low risk' if they are at term $(37+^0–41+^6$ weeks gestation), with a singleton assumed cephalic fetus, in spontaneous labour and without pregnancy or intrapartum factors identified by NICE that indicate a need for obstetric or other medical care in labour.

Women who freebirth or give birth prior to arrival at their chosen place of birth or prior to the arrival of a midwife will be excluded.

Descriptive analysis: All women giving birth at a participating NHS site between 2015 and study end.

Eligible sites: NHS maternity services using EuroKing with waterbirth facilities.

Qualitative component phase 1:

(Participants must be from the UK, either within or outside study sites).

Online discussion group participants:
► Pregnant women or who have given birth within the last 12 months.
► Midwives (all grades/positions).

► Neonatologists, obstetricians and paediatricians (including trainees).

Telephone interview participants:

► UK Neonatologists*.
► Obstetricians*.
► Paediatricians*.

*(including trainees)

Qualitative component phase 2:

Purposively selected case study sites to include midwifery and obstetric units with a range of waterbirth rates (excluding units without a waterbirth facilities).

In each case study site, participants are purposively sampled for discussions, to include the following:

► Midwives representing a range of grades and positions.
► Obstetricians, neonatologists and paediatricians (including trainees).
► Other unit staff as identified.
► Women who have given birth in the unit recently.
► Lay representatives who have experience of supporting local women who have given birth recently—for example, doulas.

### Recruitment/opt-out

For the cohort study, data will be collected on all women and babies born at participating sites from 1 January 2015 until study end date unless they choose to opt-out of the study.

The online discussion groups will be advertised via social media, magazine articles, email circulation and leaflets/flyers handed out at conferences. Adverts will provide a brief overview of the discussion groups and a website link which will contain study overview, a participant information sheet (PIS) and discussion group ground rules. If keen to participate, individuals will submit their email address via the website, which will generate an automated email invitation with a link to the discussion group registration page. Participants will be asked to complete an online consent form, create an anonymous public forum name and password to log in to the discussion, and click to confirm that they agree to comply with the discussion group ground rules.

The telephone interviews will be advertised via professional and lay networks, including social media and email circulation. Adverts will provide a brief overview of the study and the purpose of the interviews, together with a contact email address for those interested in taking part. Potential participants will be emailed a copy of the PIS and given the opportunity to ask questions. Those who would like to take part will be asked via email to agree a date for the interview.

For phase 2 of the qualitative work, data provided by sites will be used to identify midwifery-led and obstetric units, enabling the team to work with sites with both low and high waterbirth rates. Once potential sites are identified, the site Principal Investigator (PI) will be contacted by a member of the study team, provided with information about the case studies and invited to take part. For the discussions within the case study sites, the study will be publicised at unit meetings and via local networks, and potential participants requested to contact the research team to receive information. Researchers will also approach staff members and lay representatives directly to encourage voluntary participation. A PIS and opportunity to ask questions will be provided. Those who would like to take part will be asked to contact the researchers to arrange to participate in a discussion.

### Justification of approach

The POOL Study will collect data on 600 000 mother/infant dyads with three years of these data having been collected in electronic systems prior to site opening. It is not practical to ask for consent from every woman and doing so would inevitably lead to an incomplete cohort and potentially a biased sample. The study will involve the transfer of personal data to NNRD and for this we are using an opt-out model under s251, as approved by the Confidentiality Advisory Group (CAG).

### Development of the opt-out leaflet/cards

We worked closely with the study patient and public involvement (PPI) representatives on the wording of these documents. A key consideration was to ensure that there were multiple time points for the mother to opt-out of the study throughout her episode of maternity care, by informing a midwife or making contact with the maternity unit. In addition to informing all women during pregnancy that the study is running in their maternity unit, any woman who uses a pool during labour or birth will be provided with a take-home card. This will reinforce the option to opt-out of the study with relevant contact details. The final text was approved by both an NHS Research Ethics Committee (REC) and CAG committee as part of the overall governance approval for the study.

### Process to manage opt-out

If opt-out is selected, a healthcare worker will select the opt-out option on EuroKing, automatically generating a filter so that WS will not extract the mother's/infant's data, nor will data be sent to the NNRD. It was not practical to offer the chance to opt-out to women who gave birth at participating units in the period between 1 January 2015 and date of site opening.

For the online discussion groups, participants may contribute as much or as little as they wish. For face-to-face and telephone discussions, participants have the right to decline or withdraw consent at any time, without any effect on their care or employment.

### Patient and public involvement

Lay persons were involved in the original grant proposal, development of research questions, study design and outcomes. The study management group and the study steering committee have PPI representatives who were actively engaged in study design and study conduct.

## Governance and compliance

Following REC approval (18/WA/0291) and s251 support (18CAG0153) retrospective data extracts were made from the pilot site. New variables were added to EuroKing for the commencement of prospective data collection.

In order to satisfy the requirements of the s251 support the Data Security and Protection Toolkit (DSPT)[9] commissioned by the Department of Health for NHS Digital to develop and maintain) was required for both WS (registered as Healthcare Software Solutions) and NNRD (registered as Chelsea and Westminster Hospital NHS Foundation Trust). This organisation-level assessment provides reassurance of satisfactory information governance within the two organisations. Both the s251 support and DSPT are assessed and renewed on an annual basis.

## Analysis
### Sample size

The non-inferiority of birth in water compared with birth on land on rates of OASIS will be examined by parity. The Birthplace in England study found that overall 4.6% and 1.6% of nulliparous and parous women respectively, sustained OASIS.[10] A sample size of 15 000 nulliparous and 15 000 parous low-risk women (7500 each water and land) is required to obtain 90% power, and a 95% one-sided CI around a treatment difference of zero. A non-inferiority margin of 1% or less, and 0.6% or less will be taken as clinically non-significant among nulliparous and parous low-risk women, respectively. Since nulliparous women birthing in water are regarded as the least prevalent of the four groups, the data collected would provide data on 7500 would ensure adequate numbers in the other three, more prevalent groups. These data will be combined to assess the effects averaged across both strata at an increased power, with a sample size of 30 000 low-risk women. We have assumed that 25% of the 6600 waterbirths recorded in EuroKing in 2015 were nulliparous women (1650/annum). For the infant primary outcome, an estimate of 5% is used for the proportion of infants born to low-risk mothers experiencing 'adverse infant outcome or treatment'.[11] A non-inferiority margin of 1.0% or less will be taken as clinically non-significant. A sample size of 16 200 infants (8100 per group water/land) are required to have 90% power, and a 95% one-sided CI around a treatment difference of zero.

## Main analysis

Primary analysis will compare maternal and infant outcomes only between women without identified pregnancy or intrapartum complications 'low risk' who stay and give birth in the pool (waterbirth, group 1) compared with women without identified pregnancy or intrapartum complications 'low risk' who use water immersion during their labour but decide to leave the water for birth (out of water, group 2).

The primary analyses are based on a non-inferiority test of birth in water vs on land, comparing (1) the proportion

**Table 2** Potential confounders for both maternal and infant primary outcomes

| | Maternal outcome: OASIS | Adverse infant composite outcome |
|---|---|---|
| Maternal age (years) | X | X |
| Maternal BMI | X | X |
| Parity | X | |
| Duration of labour | X | X |
| Gestational age at delivery (weeks) | X | X |
| Birth weight (g) | X | X |
| Infant head circumference (cms) | X | |
| Maternal thyroid disease (including hypothyroidism) | | X |
| Prelabour ruptured membranes | | X |
| Intrapartum fever | | X |
| Small for gestational age (weight <10th centile for gestational age) | | X |
| Infant gender | | X |
| Meconium-stained liquor | | X |

BMI, body mass index; OASIS, Obstetric Anal Sphincter Injury.

of mothers that have OASIS (based on retrospective and prospective EuroKing data) and (2) the proportion of infants with a composite outcome of 'adverse infant outcome or treatment' (based on prospective EuroKing and NNRD data).

To test the primary hypothesis of non-inferiority between birth in water and on land, the maternal and infant primary outcomes will be evaluated for non-inferiority using logistic regression models, in the first instance with no adjustment for covariates. Adjusting for potential confounders may result in a more precise treatment effect estimate. The potential confounders of both primary outcomes (listed in table 2) will be considered. Directed acyclic graphs; visual representations of causal assumptions will be used to identify the presence of confounders.

The main logistic model will incorporate these selected covariates through regression adjustment. Results will be reported as an unadjusted and adjusted ORs (comparing birth in water to on land), and a two-sided 90% CI for the unadjusted and adjusted OR will be calculated. Non-inferiority will be concluded if the upper limit of the 90% CI for the difference in infant outcome between the groups is less than 1.0% (OR <1.21). Similarly, for the mother's outcome, non-inferiority will be concluded if the upper limit of the 90% CI for the difference in the proportion of OASIS between the groups is less than 1.0% (OR <1.23) in nulliparous low-risk women and less

than 0.6% (OR <1.38) in parous low-risk women. The data will then be combined to assess the effects averaged across both strata.

If non-inferiority is shown, then superiority analysis will be conducted as secondary analysis of the primary outcomes using logistic regression and will be presented as (unadjusted and adjusted) OR of outcomes in the waterbirth group compared with the birth on land group. Parameter estimates will be provided alongside 95% CI and p value. Secondary outcomes will have non-inferiority testing as detailed above.

Secondary analysis will describe the type and rates of complications among (a) low risk women who left the pool due to clinical need along with the associated maternal and infant outcomes and (b) high risk women using a pool, along with associated care (for example use of waterproof CTG cardiotocography), and maternal and infant outcomes

Maternal characteristics such as age, parity and ethnicity of all women giving birth in the study sites during data collection will be obtained and the characteristics of women who do and who not use a pool during labour, will be compared and described.

All telephone and face-to-face discussions in phase 1 and phase 2 will be audiorecorded and transcribed verbatim. For phase 1 of the qualitative component, framework analysis will be undertaken to generate key hypotheses for further exploration in phase 2. In phase 2, data will be thematically analysed initially, supported by NVivo, in order to develop an analytic framework, which will then be used to code all data.

## ETHICS AND DISSEMINATION
### Legal and ethical considerations
The cohort component uses routinely collected data without obtaining informed consent from participants; this required additional approval, s251, from a CAG. The level of national and international recognition of the importance of the smarter use of routine data, and its value to research, has never been greater. There are, however, challenges associated with using routinely collected data.[12]

One primary consideration is of unintentional identification of individuals. This risk is managed through pseudonymising identifiable data prior to matching and before being transferred to CU for analysis and data scrutiny and cleaning on arrival.

Participation in the online discussion groups and face-to-face or telephone discussions may bring back memories of difficult or distressing experiences. It is made clear in the PIS that participants can opt-out of discussions at any time, without giving a reason, and do not have to answer any questions they do not want to. There is a risk that online discussion group participants may encounter communication from other group members which causes distress. To mitigate this, participants will be asked to agree to a set of ground rules, including a section

requesting they act in a respectful way to members. The discussion will be moderated by researchers during office hours (Monday to Friday, 09:00–17:00 hours) and any unsuitable content will be removed. Repeated posting of unsuitable content by a group member will result in their being blocked. At any time, any participant who regards posted material as offensive will have the option of having the post removed from view.

There is a risk of loss of privacy for participants in the qualitative study if data which identifies them is disclosed outside the study. Any information which could identify individuals or individual workplaces will be removed following data collection and will not be used in the reporting of findings. Quotes from discussion groups or interviews may be used in reports of the research, but no individuals or individual workplaces will be identified.

### Data processors and data controllers
#### Relationship between CU and NHS sites
For the purposes of the research activities involved in POOL, CU is the data controller and the NHS site is the data processor. For the avoidance of doubt, this is not in relation to the activities carried out as part of usual clinical practice but relates to the specific use of the data made for the research and also includes the new variables added to the EuroKing system.

#### Relationship between WS, NNRD and NHS sites
WS and NNRD are data processors in respect of all NHS Data collected by them from NHS sites in the course of their normal activities and in that they are processing NHS data on behalf of NHS Sites who are data controllers.

### Dissemination of findings
Dissemination of the study results will include publication in high calibre journals through an open access agreement, a full report, a lay infographic summary aimed at

| Table 3 | List of abbreviations |
|---|---|
| **Abbreviation** | **Full details** |
| CAG | Confidentiality Advisory Group |
| CRP | C reactive protein |
| CU | Cardiff University |
| HTA | Health Technology Assessment |
| IG | Information Governance |
| NHS | National Health Service |
| NICE | National Institute for Health and Care Excellence |
| NIHR | National Institute for Health Research |
| NNRD | National Neonatal Research Database |
| OASIS | Obstetrical Anal Sphincter Injuries |
| REC | Research Ethics Committee |
| s251 | Section 251 NHS Act 2006 |
| WS | Wellbeing Software |

pregnant women and available for use by NHS providers, and distribution though social media including podcasts or similar.

In addition to dissemination of results through publication, the results will be reported to the funder, WS, NNRD and all participating sites as well as all stakeholder groups associated with the POOL Study. On completion there will be a stakeholder event for results dissemination.

Table 3 provides a list of abbreviations used throughout the paper.

**Author affiliations**
[1]Centre for Trials Research, Cardiff University, Cardiff, UK
[2]School of Healthcare Sciences, Cardiff University, Cardiff, UK
[3]Birmingham Clinical Trials Unit, University of Birmingham, Birmingham, UK
[4]School of Public Health, Imperial College London, London, UK
[5]Maternity Services, Cardiff and Vale University Health Board, Cardiff, UK
[6]Aberdeen Health Data Science Research Centre, University of Aberdeen, Aberdeen, UK
[7]PPI Representative, Cardiff, UK
[8]National Childbirth Trust and PPI Representative, London, UK
[9]Institute of Health and Society, University of Worcester, Worcester, UK

**Correction notice** This article has been corrected since it first published. The provenance and peer review statement has been included.

**Acknowledgements** The Centre for Trials Research is funded by Health and Care Research Wales and Cancer Research UK.

**Contributors** RM wrote the majority of the manuscript. SM provided content for the qualitative aspect of the manuscript. RC-J provided content for the statistical aspect of the manuscript. JS, RC-J and FVL-W cowrote the protocol. CG provided NNRD elements for the manuscript. MR, SP, CB, PB, SC, AH, BH, LM, RP and MN have all read and approved the final manuscript.

**Funding** This work was supported by the National Institute for Health Research Health Technology Assessment (project number 16/149/01).

**Disclaimer** The views and opinions expressed therein are those of the authors and do not necessarily reflect those of the NIHR HTA.

**Competing interests** None declared.

**Patient and public involvement** Patients and/or the public were involved in the design, or conduct, or reporting, or dissemination plans of this research. Refer to the Methods section for further details.

**Patient consent for publication** Not required.

**Provenance and peer review** Not commissioned; externally peer reviewed.

**ORCID iDs**
Rebecca Milton http://orcid.org/0000-0001-5985-3866
Christopher Gale http://orcid.org/0000-0003-0707-876X
Fiona V Lugg-Widger http://orcid.org/0000-0003-0029-9703
Michael Robling http://orcid.org/0000-0002-1004-036X

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
