## [Reviewer comments · BMJ Open]

ARTICLE DETAILS

TITLE (PROVISIONAL)	Establishing the safety of waterbirth for mothers and babies: A cohort study with nested qualitative component: The protocol for the POOL study.
AUTHORS	Milton, Rebecca; Sanders, Julia; Barlow, Christian; Brocklehurst, Peter; Cannings-John, Rebecca; Channon, Sue; Gale, Chris; Holmes, Abigail; Hunter, Billie; Paranjothy, Shantini; Lugg-Widger, Fiona; Milosevic, Sarah; Morantz, Leah; Plachcinski, Rachel; Nolan, Mary; Robling, Michael

VERSION 1 – REVIEW

REVIEWER	Professor David Ellwood Griffith University School of Medicine, Queensland, Australia
REVIEW RETURNED	05-Jul-2020

GENERAL COMMENTS	Thanks for allowing me to review your manuscript and please accept my sincere apologies for the delay in providing this. I am strongly supportive of this study being completed as I think the research questions(s) are of critical importance to women's birth choices, and the study design is appropriate given the limitations acknowledged.. I only have a few comments to make; 1. In the primary objective it is stated that the study will examine if water birth for low-risk women is as safe for mothers and infants, but I can't find anywhere in the manuscript that defines 'low-risk'? Also, in the secondary objectives it is stated that you will examine characteristics of, and outcomes for women with risk factors who use water for birth. How will you differentiate between women who are low and high (or increased) risk?2. I am wondering about the value of including the qualitative study nested onto the main cohort study? The research questions being addressed are quite different and it would make for a less complicated protocol of they were separated into two separate protocols.3. There appears to be an error at Line 10 on page 8 and this paragraph appears to be incomplete.
--

REVIEWER	Vanessa Scarf University of Technology Sydney, Australia
REVIEW RETURNED	06-Jul-2020

GENERAL COMMENTS	Thank you for the opportunity to review "Establishing the safety of waterbirth for mothers and babies: A cohort study with nested qualitative component: The
--

	protocol for the POOL study. This trial is seeking to answer important questions around the safety of birth in water for women (and their babies) who are clinically appropriately placed to do so. The qualitative component of the study will add interesting data around the factors that influence the use of water immersion for labour and birth. Comment on grammatical errors There are two small typos in the introduction: Page 5, line 9: after analgesia insert "be" to read 'be made available' Page 5, line 29: should say from, rather than form, after '29 UK maternity units'. Page 8, line 10: Is this a reference to Figure 1? The methodology of the protocol is sound, and the figures and tables are very informative. There is a reference to the NICE guideline for Intrapartum care for healthy women and babies, however, a clear definition of what constitutes a "low-risk" pregnancy would clarify the cohort selection process. Results from the POOL study will contribute important evidence to the use of water immersion during labour and birth.
--	---

REVIEWER	Della Forster La Trobe University, Australia
REVIEW RETURNED	09-Nov-2020

GENERAL COMMENTS	Waterbirth cohort protocol Thanks for the opportunity to review this paper. There is a clear need for more data on maternal and neonatal outcomes of waterbirth, so it is very positive the study is being undertaken. I am unclear on some aspects, and think they need further explanation and/or clarification. In more than one place it its stated there will be 'up to' 29 units will be included. I am not sure what this means, i.e. why 'up to'?. The Cohort study inclusion criteria is lacking detail and hard to understand. This is such an important aspect of a cohort study like this – the careful description of exactly who is in and out of the study. It is critical so the reader can assess risk of bias. It is quite challenging all through to get a clear understanding of exactly what the study population is. I think it is eventually fairly clear, but would be better if there was a clearer explanation earlier I think. I don't really understand what is meant by: "questions are already recorded, for example perineal trauma, therefore, data from births from 2015 onwards can be included", i.e. it is not clear on what the choice of '2015 onward' is based on. There is a lack of rationale given for the choice of primary outcome of the study. And on the composite neonatal outcome list, why is it 'all deaths' ? "or all deaths prior to NNU/postnatal ward discharge". I think it is just sentence needs fixing. Although it is understandable that type of birth cannot be an outcome as the protocol is currently written, I wonder whether the inclusion criteria should be 15,000 women planning on a water birth at the start of pushing – then this would include women and infants for whom there was some reason this did not go ahead. As it is now, there is only the opportunity for inclusion of those who actually progress to a vaginal waterbirth. Is that right? Or can the 15,000 who labour in water and then do not continue have other birth types and be included? Is there any chance that there are some women
---

	coming out where there might be a risk identified? I am not sure – but the current approach needs more explicit discussion of this I think – I just wonder if this way might end up under-estimating some of the outcomes such as the neonatal ones in particular – not things like OASIS of course. But even this could be affected if someone got out for lack of progress or ineffective pushing for example. So in the cohort main analysis, is the comparison only between those low risk women who stay and birth in the pool compared with those low risk women who simply decide to leave the water for birth, with no risk-based reason? If yes, (or if not) can this be made more explicit. So the reader can really clearly understand who is and is not in the datasets being compared. Overall there needs to be more rationale provided for a) inclusion criteria and b) outcome measures – e.g. how does having obstetric involvement in birth be justified as an outcome measure? This is likely to be an important study – so needs to be very well described and presented, with clear rationale for each aspect. This statement is not at all clear until I get to it – “The primary study aim is to compare maternal and infant outcomes for low risk women who gave birth in water (Group 1) against low risk women who left the water prior to birth for reasons other than clinical need (Group 2). The way the data will be accessed is clearly described, but not so the rationale for various aspects. What does Demographics include? How will women be informed about the study? I see they can opt out: “Participants will have the option to opt-out by informing the maternity unit that they do not wish to participate.” Is there an information leaflet available? What does “Individual sites will be facilitated...” mean? Will women be able to opt out freely? What is the process? The sample size rationale and data analysis sections have enough detail. No estimates of sample size are provided for the qualitative component.
--	--

VERSION 1 – AUTHOR RESPONSE

Comments from Reviewer 1

Comment 1: In the primary objective it is stated that the study will examine if water birth for low-risk women is as safe for mothers and infants, but I can't find anywhere in the manuscript that defines 'low-risk'? Also, in the secondary objectives it is stated that you will examine characteristics of, and outcomes for women with risk factors who use water for birth. How will you differentiate between women who are low and high (or increased) risk?

Response: Thank you for pointing this out. This was previously referenced, but not as clear as it needs to be for an international readership. We have made the following change to the manuscript in the study design section: ‘The primary study aim is to compare maternal and infant outcomes for low-risk women who gave birth in water (Group 1) against low-risk women who left the water prior to birth, with no risk-based or clinical reasons (Group 2) Figure 1 shows and details the study population groups. Women classified as low-risk for study purposes will be at term (37+0-41+6 weeks gestation), with a singleton fetus in spontaneous labour with an absence of factors that indicate that obstetric or other medical staff should have involvement in her care, or birth in an obstetric unit is advised²’ (Page 6)

Comment 2: I am wondering about the value of including the qualitative study nested onto the main cohort study? The research questions being addressed are quite different and it would make for a less complicated protocol if they were separated into two separate protocols.

Response: Thank you for this suggestion. It would have been interesting to explore this aspect. However, our study, was designed with this nested qualitative component to provide a holistic and rounded view. From the outset the funder has regarded this as a single study and for this reason we have included details of both work packages for completeness.

Comment 3: There appears to be an error at Line 10 on page 8 and this paragraph appears to be incomplete.

Response: Thank you for highlighting this, I believe the query is in relation to this section; The primary study aim is to compare maternal and infant outcomes for low-risk women who gave birth in water (Group 1) against low-risk women who left the water prior to birth for reasons other than clinical need (Group 2) Figure 1 shows the study population groups. I will fix the link to Figure 1.

Comments from Reviewer 2

Comment 1: There are two small typos in the introduction:

Page 5, line 9: after analgesia insert "be" to read 'be made available'

Page 5, line 29: should say from, rather than form, after '29 UK maternity units'.

Page 8, line 10: Is this a reference to Figure 1?

Response: Thank you for taking the time to review this manuscript and your comments, I have amended all typos.

Comments from Reviewer 3

Comment 1: In more than one place it is stated there will be 'up to' 29 units will be included. I am not sure what this means, i.e. why 'up to'?

Response: Thank you for your comment, at the time of writing this protocol paper we were unsure of exactly how many NHS sites we would be able to engage with on the study due to contract cessation with Wellbeing Software for a few NHS sites, and others starting to use the system. This has now been finalised and the total number is 26. This has been changed throughout the manuscript.

Comment 2: The Cohort study inclusion criteria is lacking detail and hard to understand. This is such an important aspect of a cohort study like this – the careful description of exactly who is in and out of the study. It is critical so the reader can assess risk of bias. It is quite challenging all through to get a clear understanding of exactly what the study population is. I think it is eventually fairly clear, but would be better if there was a clearer explanation earlier I think.

Response: Thank you for your comment, the cohort study will include all women who used a birth pool or bath for water immersion during established labour between 1st January 2015 and the end of data collection. Women who freebirth or give birth prior to arrival at their chosen place of birth or prior to the arrival of a midwife will be excluded. Women will be classified as 'low risk' if they are at term (37+0 – 41+6 weeks gestation), with a singleton assumed cephalic fetus, in spontaneous labour, and without pregnancy or intrapartum factors identified by NICE that indicate a need for obstetric or other medical care in labour. This has now been added to the inclusion criteria (page 10).

Figure 1 defines the groups of women that will be identified within the study, this has been updated as has the analysis section.

Primary analysis will compare maternal and neonatal outcomes only between women without identified pregnancy or intrapartum complications 'low-risk' who stay and give birth in the pool (waterbirth, Group 1) compared with women without identified pregnancy or intrapartum complications 'low-risk' who use water immersion during their labour but decide to leave the water for birth (out of water, Group 2). Women who were considered to be without complications and remained in the pool intending to give birth in water, but who at birth experienced a shoulder dystocia or previously unrecognised breech presentation, with the baby then partially being born into water, (including for example only head, legs or buttocks) will continue to be included in Group 1. Similarly, women in Group 2 will include women who were regarded as 'low-risk' but who subsequently experienced shoulder dystocia or breech presentation identified at birth.

Group 3 will include women who were identified by their midwife as not having any risk factors when they first got in the pool but later developed, or had a complication in labour identified, and left the water prior to birth. Secondary analysis will describe the type and rates of complications in Group 3 along with the associated maternal and neonatal outcomes.

Group 4 will include women who were known to have risk factors when they first got in the pool. Secondary analysis for Group 4 will describe the type and rates of known risk factors among women using a pool, along with associated care (for example use of waterproof CTG) maternal and neonatal outcomes.

Maternal characteristics such as age, parity and ethnicity of all women giving birth in the study sites during data collection will be obtained and the characteristics of women who do and who not use a pool during labour, will be compared and described.

Comment 3: I don't really understand what is meant by: "questions are already recorded, for example perineal trauma, therefore, data from births from 2015 onwards can be included", i.e. it is not clear on what the choice of '2015 onward' is based on.

Response: This section has now been re-written to hopefully a more coherent sentence within the study design section (page 5): A larger sample size is required for the maternal (30,000) compared to the neonatal (16,200) primary outcome. To inform the maternal primary outcome, severe perineal trauma, which is already collected in the maternity information system at study sites, data will be extracted relating to births between January 1st, 2015 to the end of data collection. The neonatal composite primary outcome includes data items added at site opening, as essential data are not collected routinely. For this reason, data used to inform the neonatal primary outcome will only include births occurring between the date of an individual site opening and the end of data collection.

Comment 4: There is a lack of rationale given for the choice of primary outcome of the study. And on the composite neonatal outcome list, why is it 'all deaths' ? "or all deaths prior to NNU/postnatal ward discharge". I think it is just sentence needs fixing.

Response: Thank you for this comment, you're correct this sentence needs fixing. All statements relating to death now state: "intrapartum stillbirth or deaths prior to NNU/postnatal ward discharge" throughout the document.

Comment 5: Although it is understandable that type of birth cannot be an outcome as the protocol is currently written, I wonder whether the inclusion criteria should be 15,000 women planning on a water

birth at the start of pushing – then this would include women and infants for whom there was some reason this did not go ahead. As it is now, there is only the opportunity for inclusion of those who actually progress to a vaginal waterbirth. Is that right? Or can the 15,000 who labour in water and then do not continue have other birth types and be included? Is there any chance that there are some women coming out where there might be a risk identified? I am not sure – but the current approach needs more explicit discussion of this I think – I just wonder if this way might end up under-estimating some of the outcomes such as the neonatal ones in particular – not things like OASIS of course. But even this could be affected if someone got out for lack of progress or ineffective pushing for example. So in the cohort main analysis, is the comparison only between those low risk women who stay and birth in the pool compared with those low risk women who simply decide to leave the water for birth, with no risk-based reason? If yes, (or if not) can this be made more explicit. So the reader can really clearly understand who is and is not in the datasets being compared.

Response: Many thanks for your succinct summary at the end of this paragraph. The existing paragraph in Study Design (page 6) has been altered to read: The primary study aim is to compare maternal and infant outcomes for low-risk women who gave birth in water (Group 1) against low-risk women who left the water prior to birth, with no risk-based or clinical reasons (Group 2) Figure 1 shows the study population groups.

Comment 6: Overall there needs to be more rationale provided for a) inclusion criteria and b) outcome measures – e.g., how does having obstetric involvement in birth be justified as an outcome measure?

Response: Thank you for your comment, it is a valid and constructive point. The POOL Study is about describing the whole population of women using a pool and is an opportunity to describe outcomes for women that use a pool but also require some obstetric care during labour. We have amended the study description to make this clearer.

Comment 7: This is likely to be an important study – so needs to be very well described and presented, with clear rationale for each aspect. This statement is not at all clear until I get to it – “The primary study aim is to compare maternal and infant outcomes for low-risk women who gave birth in water (Group 1) against low-risk women who left the water prior to birth for reasons other than clinical need (Group 2).

Response: I have altered this sentence in accordance with your comment 5 and other similar comments from the other reviewers. The section on study design has been substantially amended and we hope this is now clearer.

Comment 8: The way the data will be accessed is clearly described, but not so the rationale for various aspects. What does Demographics include?

Response: The demographics that will be used to explore and compare the pre identified groups in the study include parity, age, ethnicity and deprivation (Table 1 has been updated, page 8)

Comment 9: How will women be informed about the study? I see they can opt out: “Participants will have the option to opt-out by informing the maternity unit that they do not wish to participate.” Is there an information leaflet available? What does “Individual sites will be facilitated...” mean? Will women be able to opt out freely? What is the process?

Response: The paragraph on ‘Opportunity to Opt-Out’ (page 10) has been updated to include details of the Patient Information Sheet, posters and cards: Participants will have the option to opt-out by informing the maternity unit that they do not wish to participate. CU will provide individual sites with patient information leaflets, posters and take-home cards. Individual sites are responsible for ensuring

all women are provided with relevant information to ensure they are aware of the option not to have their data included in the study.

Comment 10: The sample size rationale and data analysis sections have enough detail. No estimates of sample size are provided for the qualitative component.

Response: Thank you for your comment on the sample size aspect of the qualitative work. The qualitative component used a case sampling approach based on the key dimensions of type of unit, geographical diversity and differing waterbirth rates.

Due to the additional detail required to make this manuscript clearer we are now slightly over the word count at 4,316. I do hope that the improvements made outweigh the additional words, however if you would like us to reduce the word count please do advise. We would like to again thank you and the three reviewers for your comments and input into improving this manuscript.

VERSION 2 – REVIEW

REVIEWER	David Ellwood Griffith University School of Medicine, Queensland, Australia
REVIEW RETURNED	06-Dec-2020
GENERAL COMMENTS	Thank you for addressing my comments and those of the other reviewers. I have no further comments or questions.
REVIEWER	Vanessa Scarf University of Technology Sydney
REVIEW RETURNED	04-Dec-2020
GENERAL COMMENTS	The revised protocol has answered the reviewers questions and addressed the concerns.
REVIEWER	Della Forster La Trobe University Australia
REVIEW RETURNED	14-Dec-2020
GENERAL COMMENTS	Thanks for the clarifications provided, and for the changes to the protocol to increase the reader's understanding of exactly who is in and out of the study etc .